# Astaxanthin, a Marine Carotenoid, Maintains the Tolerance and Integrity of Adipose Tissue and Contributes to Its Healthy Functions

**DOI:** 10.3390/nu13124374

**Published:** 2021-12-06

**Authors:** Allah Nawaz, Yasuhiro Nishida, Akiko Takikawa, Shiho Fujisaka, Tomonobu Kado, Aminuddin Aminuddin, Muhammad Bilal, Ishtiaq Jeelani, Muhammad Rahil Aslam, Ayumi Nishimura, Takahide Kuwano, Yoshiyuki Watanabe, Yoshiko Igarashi, Keisuke Okabe, Saeed Ahmed, Azhar Manzoor, Isao Usui, Kunimasa Yagi, Takashi Nakagawa, Kazuyuki Tobe

**Affiliations:** 1Department of Molecular and Medical Pharmacology, Faculty of Medicine, University of Toyama, 2630 Sugitani, Toyama 930-0194, Japan; ishwish66@gmail.com (I.J.); ksuke71@med.u-toyama.ac.jp (K.O.); nakagawa@med.u-toyama.ac.jp (T.N.); 2First Department of Internal Medicine, Faculty of Medicine, University of Toyama, 2630 Sugitani, Toyama 930-0194, Japan; octopacy1978@gmail.com (Y.N.); takikawa@med.u-toyama.ac.jp (A.T.); shihof@med.u-toyama.ac.jp (S.F.); dr.asaasa@gmail.com (T.K.); aminuddin@med.unhas.ac.id (A.A.); drbilal@med.u-toyama.ac.jp (M.B.); rahil.aslam17@gmail.com (M.R.A.); anakashi@med.u-toyama.ac.jp (A.N.); tkuwano@med.u-toyama.ac.jp (T.K.); yoshiw@med.u-toyama.ac.jp (Y.W.); yoshiko@med.u-toyama.ac.jp (Y.I.); yagikuni@med.u-toyama.ac.jp (K.Y.); 3Fuji Chemical Industries, Co., Ltd., 55 Yokohoonji, Kamiich-machi, Nakaniikawa-gun, Toyama 930-0405, Japan; 4Department of Nutrition, Faculty of Medicine, University of Hasanuddin, Makassar 90245, Indonesia; 5Center for Clinical Research, Faculty of Medicine, Toyama University Hospital, University of Toyama, 2630 Sugitani, Toyama 930-0194, Japan; 6Department of Medicine and Surgery, Rawalpindi Medical University, Rawalpindi 46000, Pakistan; saeedchachar987654@gmail.com; 7Indus Hospital Bhong, Sadiqabad 64350, Pakistan; aliimrankhan255@gmail.com; 8Department of Endocrinology and Metabolism, Dokkyo Medical University, Mibu 321-0293, Japan; isaousui@dokkyomed.ac.jp

**Keywords:** Astaxanthin, natural antioxidant, obesity, insulin resistance, adipose tissue remodeling, adipose tissue macrophages

## Abstract

Recently, obesity-induced insulin resistance, type 2 diabetes, and cardiovascular disease have become major social problems. We have previously shown that Astaxanthin (AX), which is a natural antioxidant, significantly ameliorates obesity-induced glucose intolerance and insulin resistance. It is well known that AX is a strong lipophilic antioxidant and has been shown to be beneficial for acute inflammation. However, the actual effects of AX on chronic inflammation in adipose tissue (AT) remain unclear. To observe the effects of AX on AT functions in obese mice, we fed six-week-old male C57BL/6J on high-fat-diet (HFD) supplemented with or without 0.02% of AX for 24 weeks. We determined the effect of AX at 10 and 24 weeks of HFD with or without AX on various parameters including insulin sensitivity, glucose tolerance, inflammation, and mitochondrial function in AT. We found that AX significantly reduced oxidative stress and macrophage infiltration into AT, as well as maintaining healthy AT function. Furthermore, AX prevented pathological AT remodeling probably caused by hypoxia in AT. Collectively, AX treatment exerted anti-inflammatory effects via its antioxidant activity in AT, maintained the vascular structure of AT and preserved the stem cells and progenitor’s niche, and enhanced anti-inflammatory hypoxia induction factor-2α-dominant hypoxic response. Through these mechanisms of action, it prevented the pathological remodeling of AT and maintained its integrity.

## 1. Introduction

Obesity is not only a phenomenon of weight gain, but it is also widely known to be associated with various diseases. The increasing prevalence of obesity and associated comorbidities such as type 2 diabetes, cardiovascular disease, and certain cancers represents a major threat to public health [1]. The total number of people with diabetes is estimated to rise from 451 million today to 693 million by 2045 [2], and the prevalence of type 2 diabetes is expected to rise in children and adolescents around the world in all ethnicities [3]. In 2017, about 5 million deaths were attributed to diabetes [2]. These alarming statistics highlight that only a few effective treatment strategies exist to fight this multifactorial disease. Insulin resistance precedes and predicts type 2 diabetes and it underlies the development of many comorbidities. Two major lifestyle changes, namely decreased physical energy expenditure and the increased availability and abundance of palatable high-fat foods, have contributed to the development of obesity and insulin resistance. One potential mechanism of insulin resistance involves cellular oxidative stress induced by excess energy states. Adipose tissue (AT) is not only a pool of fat, but also an endocrine tissue, which is essential for the regulation of systemic metabolic functions and the development of various diseases through the production of many cytokines, called adipokines, and the release of free fatty acids [4,5,6].

In recent years, a variety of nutrients have been reported to exert direct effects on AT, but the mechanism of action of most of them is either to inhibit the development of adipocytes or to lipolysis of fat from adipocytes. In other words, the goal is simply to provide weight loss, but it is unclear whether these actions always produce a physiologically beneficial effect [7,8].

We have shown that astaxanthin (AX) (Figure 1), a marine carotenoid known as a strong antioxidant, ameliorates insulin resistance in diet-induced obese mice or in vitro myotubes by modulating insulin signaling in an antioxidant activity-dependent and -independent manner and by activating mitochondrial energy metabolism via activating the AMP-activated protein kinase (AMPK)/peroxisome proliferator activated receptor γ coactivator-1α (PGC-1α) pathway in skeletal muscle [9,10]. These findings focused mainly on skeletal muscle, but on the other hand, the possibility of exerting anti-inflammatory effects in AT has also been partially reported in animal models [10,11,12,13,14] and humans [15,16,17,18]. According to these reports, the effects of AX are thought to be mainly based on its antioxidant activity, with increased HDL and increased adiponectin levels in the blood. The role of AX in AT might be to suppress chronic inflammation and maintain non-alcoholic steatohepatitis and pancreatic functions by inhibiting obesity-induced inflammatory M1 macrophage (MΦ) infiltration, the production of pro-inflammatory cytokines, and the release of free fatty acids. However, it remains unclear how AX works on adipocytes and stromal vascular fraction (SVF) in AT. In this study, we aimed to directly and comprehensively elucidate the effects of AX on AT.

## 2. Materials and Methods

### 2.1. Reagents

The cell culture reagents were purchased from Invitrogen (Carlsbad, CA, USA); commercially available astaxanthin (AX) powder was purchased from Fuji Chemical Industries USA; Inc. (product name; P2AF, containing 2% of AX from *Haematococcus pluvialis*, Burlington, NJ, USA); All the other reagents were purchased from Sigma-Aldrich (St Louis, MO, USA).

### 2.2. Animals

Five-week-old male C57BL/6J mice were purchased from Sankyo Laboratory Service (Tokyo, Japan). All the animals were housed in a 12 h light/12 h dark cycle and allowed free access to food and water. The regular diet (normal chaw (NC); D12450B) and a 60% high-fat diet (HFD; D12492) and their AX pre-mixed diet (final AX content was 0.02% using commercially available AX powder) were purchased from Research Diets Inc., (New Brunswick, NJ, USA). One week after habitation, from 6 weeks of age, they were fed on each of these diets. Ten or twenty-four weeks after the administration of HFD, the mice were sacrificed after anesthesia to harvest tissues for analysis. The animal care policies and procedures for the experiments were approved by the animal experiment committee at the University of Toyama.

### 2.3. Body Composition Analysis

Body composition, including fat and lean mass, was compared using EchoMRI-100 (Hitachi Aloka, Hitachi, Japan) at ten or twenty four weeks after administration. After fasting, the animals were sacrificed after anesthesia through intraperitoneal injection, blood was collected and liver and epididymal white adipose tissue were removed and their weights measured. The collected tissues were also used for the following miscellaneous tests.

### 2.4. Realtime Reverse Transcription-Polymerase Chain Reaction (Realtime RT-PCR)

Tissues for the RT-PCR were collected and preserved in an RNA solution from Ambion (Austin, TX, USA) according to the manufacturer’s instructions. All the tissues, except for the skeletal muscle, were lysed in a buffer RLT in an RNeasy kit (Qiagen, Hilden, Germany). For the skeletal muscle, approximately 100 mg of tissue was minced and lysed with 1 mL of Isogen (Nippon gene, Toyama, Japan), before the addition of 200 μL of chloroform. After centrifugation, the transparent/clear layer was transferred to a new Eppendorf tube. The RNA extraction and RT-PCR were performed as previously described [6,10,19,20].

### 2.5. Glucose Tolerance Test and Insulin Tolerance Test

In the intraperitoneal glucose tolerance test (IP-GTT), 18 h fasted mice were injected with glucose (1 mg/g BW) intraperitoneally. In the intraperitoneal insulin tolerance test (IP-ITT), 2 h fasted mice were injected intraperitoneally with human insulin 0.8 units/kg BW for NC-fed and 1.2 units/kg BW for HFD-fed mice. Blood samples were then collected after 0, 15, 30, 45, 60, 90, and 120 min from the tail vein. The blood glucose levels were measured using the STAT STRIP Express 900 (Nova Biomedical, Waltham, MA, USA).

### 2.6. Flow Cytometry Analysis

The isolation and separation of the SVF and subsequent flow cytometry were per-formed as described previously [20], with minor modification. Briefly, negative selection of 7AAD (live cells) were gated for CD45^+^ (hematopoietic) cells, followed by the positive selection of F4/80^+^ cells. Next, F4/80^+^ cells were gated for CD11c^+^ and CD206^+^ cells. This experiment was performed using Becton Dickinson FACS Canto II and the data were analyzed using FlowJo (V10) software.

### 2.7. Immunohistochemistry

Paraffin-embedded tissues were cut 5 μm thick and mounted on slides. Slide staining with hematoxylin and eosin (H&E) was performed according to the manufacturer’s instructions. A specific cell count was performed for at least three randomly chosen 200× magnification fields. The average adipocyte size was analyzed by NIH Image J software at least three randomly chosen.

### 2.8. Oxidative Stress Analysis

The oxidative stress were evaluated by production of malondialdehyde (MDA) and thiobarbituric acid-reactive substances (TBARS) in frozen tissues. TBARS assays were measured using a “Malondialdehyde (MDA) Assay Kit” (Product No. NWK-MDA01, Northwest Life Science Specialties, LLC., Vancouver, WA, USA), according to the manufacturer’s instructions.

### 2.9. Statistical Analysis

The Statistical analyses were performed using unpaired Student’s *t*-tests or One- or Two-way ANOVAs with the Dunnett’s or Tukey-Kramer posttest. The differences were considered statistically significant at * *p* < 0.05, ** *p* < 0.01 and *** *p* < 0.001. The results are presented as the means ± SEM.

## 3. Results

### 3.1. AX Inhibits Increases in Liver Weight, but Increases the Weight of Adipose Tissue upon High-Fat Diet Stimuli

We previously reported that astaxanthin (AX) supplementation mixed with a high-fat diet (HFD) prevents the progression of the early stages of weight gain associated with feeding, but we did not observe any difference in weight gain in later stages compared to the control group (24 weeks of HFD-feeding) [10]). However, when we stratified the data between the early and late periods of rearing, there were significant changes in weight gain. This means that weight gain was suppressed in the AX-treated group in the early period (0 to 8 weeks after treatment) and, conversely, increased in the late period (16 to 24 weeks after treatment) (Appendix A). In addition, as the HFD treatment period prolonged to 76 weeks, the AX-treated HFD group gained more body weight compared to the HFD control group (Appendix A). However, AX administration significantly prevented the deterioration of blood glucose, blood lipids, and blood pressure [10]. At this point, computed tomography imaging showed unbiased increases in both visceral fat and subcutaneous fat (data not shown). The Fat/lean ratio evaluated by MRI was significantly lower in the AX-treated group at 8 weeks, whereas the ratio was higher in the AX-treated group at 24 weeks, although it was not significant (Figure 2A). Consistent with these results, the weight of epididymal white adipose tissue (eWAT) was significantly lower in the AX-treated group at 10 weeks, but was reversed at 24 weeks of HFD-feeding, with a significantly higher weight in the AX-treated group (Figure 2B). When evaluated for liver weight, the AX-treated group showed significantly lower weight compared with the control group at 24 weeks of HFD-feeding (Figure 2C). The control group demonstrated a decrease in eWAT weight and an increase in liver weight at 24 weeks compared to 8 weeks of HFD-feeding. The AX group exhibited a smaller magnitude of this change. Therefore, it was considered that the AX treatment prevented ectopic fat accumulation in the liver by correctly depositing excess fat of dietary origin in adipose tissue (AT). Furthermore, the reduction in the levels of lipids in the blood when compared to the HFD control group was suggested to be the effect of the prevention of liver dysfunction by preventing the accumulation of ectopic fats, such as steatosis, via AT integrity, in addition to the improvement of the lipid utilization function of skeletal muscle by AX administration [10]. Interestingly, the eWAT and liver in the AX-treated group appeared to be orange in color (Figure 2D). Similar findings were also observed in other Ats, such as inguinal white adipose tissue (iWAT) and brown adipose tissue (BAT) (data not shown). This suggests that AX administration results in gross morphological changes in AT, including pigmentation.

### 3.2. AX Decreases Oxidative Stress in Adipose Tissue and Suppresses the Expression of Pro-Inflammatory Cytokine Genes

In a previous study, we demonstrated that AX treatment only reduces oxidative stress measured by 2-thiobarbituric acid reactive substances (TBARS) in eWAT [10]). According to the gene expression analysis of antioxidant enzymes and their regulators in these tissues, the AX-treated group exhibited downregulated expression levels of the oxidative stress marker gene (*Hmox-1*)*,* and upregulated expression levels of *Sirt3* and *Catalase* compared to the control HFD group, but no systematic changes in the expression levels of other antioxidant enzymes, including *Nrf2,* were observed between the groups (Appendix A). Interestingly, we found downregulated expressions of pro-inflammatory cytokines regardless of the time period of the study (Figure 3A,B, and Appendix A). This change was more significant in the 24 week treated mice. For example, in the AX-treated group, there were decreased gene expression levels of pro-inflammatory cytokines, and chemokines such as *Ccr2*, *Il-1β*, *Il-6*, *Nlrp3*, *Nos2*, and *Tnf*α, and increased gene expression levels of the anti-inflammatory cytokine *Il-10* were observed. This suggests that the AX-treated group had reduced inflammation in the eWAT due to the antioxidant activity of AX itself or some other mechanisms. Notably, the AX-treated group featured higher gene expression levels of peroxisome proliferator-activated receptor γ (PPARγ), which is a nuclear receptor and a target of anti-diabetic drugs such as thiazolidinediones; it is expressed in smaller adipocytes that secrete more metabolically favorable adipokines [21]. AX has been also reported to be a partial modulator of PPARγ [22]. As for adiponectin (*AdipoQ*), which is a gene transcriptionally regulated by PPARγ [21], and a favorable adipokine secreted from AT, it increased in the AX-treated group regardless of the treatment period (Figure 3C, and Appendix A), which reflects the effects that were observed in humans and obese animals [7,11,12].

The activation of PPARγ, in association with peroxisome proliferator activates receptor γ coactivator-1α (PGC-1α), induces adipocyte progenitor cell differentiation to “Beige/Brite (brown-like in white)” adipocytes cells with high expression of uncoupling protein 1 (UCP-1) located in the inner mitochondrial membrane [23,24]. UCP1 is responsible for uncoupling the pathway of oxidative phosphorylation for ATP synthesis and non-shivering thermogenesis by dissipating chemical energy as heat and promoting high levels of fatty acid oxidation. We have also previously shown that AX induces AMPK-mediated gene expression of *Pgc-1a* in skeletal muscle [10]. Since the gene expression of *Ucp-1* seemed to upregulate in the eWAT of mice treated with AX (Figure 3C, and Appendix A), we evaluated the AT beigeing phenomenon in the inguinal or subcutaneous white adipose tissue (iWAT). In the iWAT of mice treated with AX for 8 weeks, the expression of *Ucp-1* and beige/brite adipocyte marker genes were significantly upregulated compared with the non-AX-treated group (Appendix A). In these animals, the gene expression of *Pgc-1α*, but not *Prdm16*, was upregulated, suggesting that these changes were due to the effects of PPARγ and PGC-1α. Unfortunately, however, the glucose tolerance and insulin sensitivity of these animals were similar regardless of the strength of *Ucp-1* expression in iWAT, and there was no extreme enhancement of *Ucp-1* expression in iWAT at week 24 (data not shown). Therefore, it was considered that this phenomenon had minor effects on the ameliorating effects of AX treatment on AT inflammation and whole body energy metabolisms. Therefore, we hypothesized that the central role of AX in AT is its anti-inflammatory effect, and focused our research there.

It is well known that the production of inflammatory cytokines in AT is associated with the accumulation of inflammatory M1 macrophages (MΦ) in AT due to damage-associated molecular patterns (DAMPs) caused by oxidative stress and hypoxia-induced adipocyte damage and cell death [20]. These conditions also led to the downregulated expressions of anti-inflammatory M2 MΦ, which are tissue-resident and/or regulatory MΦ. We also reported that M2 MΦ play an important role in maintaining the functions and homeostasis of AT [6]. Therefore, we evaluated the effects of AX on macrophages in AT. In the AX-treated group, the gene expression of F4/80, a surface marker of MΦ in mice, was decreased in eWAT, which may have reduced MΦ infiltration in AT (Figure 3A,B). Remarkably, the gene expression of *Cd11c*, one of the M1 MΦ markers, was significantly decreased, although the gene expression of *Cd206*, one of the markers of AT M2 MΦ, remained unchanged. Therefore, we next evaluated the quantity and quality of MΦ in the eWAT by FACS to further clarify the inflammation.

Consistent with gene expression analysis, our flow cytometry analysis also revealed that AX treatment significantly reduced the number of CD11c+ M1 MΦ, while the number of CD206+ M2 MΦ remained unchanged (Figure 4A). Interestingly, the M1/M2 MΦ ratio, which was increased by HFD feeding, was significantly decreased in the AX-treated group, mainly due to a significant decrease in the accumulation of M1 MΦ in eWAT, and no significant change in the number of M2 MΦ (Figure 4A–C). Therefore, consistent with the results of the gene expression, AX treatment prevented the accumulation of M1 MΦ in eWAT. Consistent with previous reports [14], our data also showed that AX treatment did not alter the polarization of MΦ. From these results, we concluded that the anti-inflammatory effect of AX in AT was mainly due to the reduced accumulation of M1 MΦ.

Next, we investigated the size of the adipocytes in eWAT for evaluating the function of AT. Excessively hypertrophic adipocytes can cause cytotoxicity due to hypoxia, leading to chronic inflammation. On the other hand, smaller adipocytes secrete beneficial adipokines, such as adiponectin, and are able to store excess fat. We evaluated the impact of AX on them. In brief, the diameter was evaluated based on the calculated area of individual adipocytes stained with Hematoxylin and Eosin (H&E) (Figure 5A). We did not observe any difference in the number of hypertrophic adipocytes larger than 100 μm in the AX-treated group compared to the HFD control group. However, there was a significantly higher number of smaller adipocytes of approximately 20 μm in the AX-treated group.

Hypertrophic adipocytes induce a vicious cycle that is detrimental to insulin resistance and chronic inflammation, as discussed above. The microenvironment that represents these events is a crown-like structure (CLS) that is highly stained with H&E, where MΦ accumulate around dead cells. We hypothesized that CLS would be significantly diminished in the eWAT of AX-treated mice; we counted CLS+ numbers (Figure 5B), and found that number of CLS was comparable between the AX-treated group and the control HFD group. This result was contradicted by the results of the FACS analysis and gene expression analysis in eWAT. Therefore, we considered that the stromal structure of eWAT in the AX-treated group was different from that of CLS, and further gene expression analysis was performed on the SVF used for FACS (Figure 6A,B).

### 3.3. AX Reduces Inflammation of SVF in eWAT and Maintain Pool Size of Adipocyte Progenitors

To evaluate gene expression in the SVFs from eWATs used in the study, we first evaluated gene expression of series of M1/M2 MΦ associated pro/anti-inflammatory markers. Consistent with the results of the FACS and gene expression in eWAT, gene expression in SVF revealed the downregulation of the gene expression of M1 macrophage-associated pro-inflammatory markers (Appendix A). In contrast to the eWAT, the gene expression of not only *F4/80* and *Cd11c*, but also *Cd206* in the AX-treated group was downregulated in SVF; similarly, the expression of anti-inflammatory marker genes was also downregulated. By contrast, the expression of *Foxp3*, a regulatory CD4+ T cell marker, was upregulated. It was speculated that this result was due to the reduced number of macrophages in the total SVF (Appendix A). Note, the expression of oxidative stress marker gene *Hmox-1* and the oxidative stress response gene in SVF were also downregulated in the AX-treated group compared to the control group.

To learn which cells in SVF were affected in the AX-treated group, we examined the gene expression of vascular marker genes, including endothelial cells and pericytes, and also evaluated the expression of adipocyte progenitors (APs) and mesenchymal stem cell (MSC)-related marker genes in eWAT and SVF. Interestingly, the AX-treated HFD group showed upregulated expression of vascular marker genes, including *VE-cadherin Cd31*, *Kdr*, *Flt1*, *Ng2*, *Nos3*, and vascular endothelial growth factors (VEGFs) marker genes including *Vegfb*, *Vegfa120*, *Vegfa164*, *Angptl4*, and *Fgf1* in eWAT compared to the control HFD group (Figure 6A). These genes, except for *Vegfs*, and *Fgf1*, were more distinctly and significantly upregulated in the SVF of the AX-treated HFD group (Figure 6B), indicating that AX promotes angiogenesis and the healthy expansion of AT. Therefore, it is possible that there was more angiogenesis or vasculogenesis occurring in the eWAT of AX-treated mice.

Maintaining niches for MSCs and APs and the healthy differentiation of adipocytes are essential to maintain stable vascular structure and angiogenesis [6,25,26]. Therefore, we hypothesized that the AX-treated mice would display more preserved MSCs and APs, and we evaluated these markers. As a result, the gene expression of a series of MSC-related marker genes, including *Pdgfrb*, *Cd90*, and *Cd105*, APs-related marker genes including *Sca-1*, *Pdgfra*, and *Pref-1*, and an early adipogenesis marker, *Cebpa* in SVF, were upregulated in the AX-treated group compared to the control HFD group (Figure 7). Therefore, it was considered that most of the cells comprising the SVF were not due to the infiltration of exogenous M1 macrophages, but rather to endogenous groups of cells comprising vasculature, MSCs and APs. Based on these results, the CLS observed in the eWAT of the AX-treated group may be the regions where adipogenesis occurred in and around the vasculatures, which were distinct from conventional CLS associated with cell death and macrophage accumulation

### 3.4. AX Induces a Predominantly HIF-2α, but Not HIF-1α, Hypoxic Response in eWAT

Based on the results suggesting more angiogenesis in the AX-treated group, indicating the microenvironment in eWAT was altered by AX administration. In the inflammation of AT, the hypoxic response due to hypoxia in their MΦ caused by hypertrophic adipocytes plays an important role in the development of insulin resistance in obese mice [20]. Under hypoxic conditions, the hypoxic response of each cell involves Hypoxia-Induction Factor (HIF)-mediated gene expressions, but the response varies between different isoforms of HIF. Interestingly, a great number of factors influences the hypoxic response, including oxygen tension, metabolites, and ROS [27,28]. Next, we examined the impact of AX on the hypoxic response of eWAT and SVF through gene expression analysis, and found downregulated expression of hypoxia-related marker genes, including *Hif1a*, *Spp1*, *Egnl1* etc., in the eWAT and SVF of AX-treated group compared with non-AX-treated HFD group, in contrast, *Epas1* (HIF-2α) gene expression was upregulated (Figure 8A,B). In contrast to HIF-1α, the HIF-2α-dominant hypoxic response functions in the beneficial manner for healthy adipose hypertrophy [20,29,30]. These findings suggest that AX treatment reduces AT hypoxia and facilitates the healthy expansion of WAT.

### 3.5. Intervention with AX in Mice after the Completion of Pathological Obesity Improved Insulin Resistance and Glucose Intolerance

Finally, most of the previous studies of AX administration to obese animals reported preventive administration even before pathogenesis caused by obesity. Therefore, we conducted an intervention study to evaluate the efficacy of AX in mice with pathological conditions such as insulin resistance and glucose intolerance caused by HFD-feeding. The animals fed a HFD for 12 weeks significantly increased their body weight compared to mice (Appendix A). These mice were randomly divided into two groups: one group received a HFD mixed with 0.02% AX, and another group continued to feed on the HFD alone (Appendix A). This time point was used as a baseline, and the glucose tolerance, insulin sensitivity, and systolic blood pressure of each mouse were evaluated (Appendix A). Since baseline glucose tolerance, insulin sensitivity, and blood pressure were significantly impaired compared to the lean mice [10], we concluded that the obesity-induced pathological insulin resistance-related conditions were sufficiently established. The treatment was then continued for another 12 weeks, and glucose tolerance, insulin sensitivity, and blood pressure were compared to baseline (Appendix A). First, there was no significant change in weight gain with or without AX administration (Appendix A). In regard to glucose tolerance, there was a significant difference in the AX group compared to the HFD control group, and also improved from baseline (Appendix A). For systolic blood pressure and insulin sensitivity, there was no deterioration from baseline, but there were no significant improvements from baseline (Appendix A). These results suggest that interventions in AX obesity should be initiated as early as possible, and may be less potent for obesity that is already established completely.

## 4. Discussion

The current study revealed that AX administration suppresses liver fat, but modest body weight gain and AT weight gain was still observed in chronic obese states. These results suggested that AX treatment prevented the ectopic accumulation of fat in the liver and other organs because excess fat of dietary origin was correctly stored in AT. The release of excess saturated free fatty acids and pro-inflammatory cytokines from AT induces insulin resistance in peripheral and central systems as well as leptin resistance in the hypothalamus [31]. In addition, healthy AT is an endocrine organ that secretes substances called adipokines, which also possess beneficial metabolic features, such as adiponectin and leptin, contributing to the maintenance of homeostasis in glucose and lipid metabolism in the whole body [32]. It is important for the benign expansion of fat deposition in response to excess calories to possess efficient storage and release of fatty acids in adipocytes and sufficient extensibility of the fat-vascular network [29].

In the eWAT of the AX-treated animals, we observed significant upregulated expression of vascular and neovascularization-related marker genes. Furthermore, the decrease in the expression of MSCs and APs-related marker genes in SVF due to excessive obesity was significantly prevented by AX treatment. This means that AX preserves and sustains MSC and APs niche in AT. Thus, it may exert a beneficial preventive effect on lipodystrophic diabetes caused by the exhaustion of progenitor cells and stem cells.

Previous studies have clearly demonstrated that adipogenesis is an important factor to maintain the metabolic health of tissue, and the imbalance between cellular hypertrophy and hyperplasia in AT depots is associated with metabolic disturbances [4,5,6,7,8,9,10,11,12]. Obesity caused by the expansion of AT through adipogenesis is considered more metabolically favorable, as it is associated with preserved insulin sensitivity, while a greater degree of hypertrophy with less hyperplasia is associated with metabolic disorders [4,6,13,14,15,16]. We recently reported that APs contribute to adipogenesis during the progression of obesity [33] and M2 macrophages enhance adipogenesis through the recruitment of APs to preserver systemic insulin sensitivity [6]. Similarly, AX treatment enhances adipogenesis and induces the expression of metabolically favorable genes, including *Pparγ* and adiponectin, suggesting that AX treatment results in the healthy metabolic adaptation of AT.

In the eWAT of AX-treated mice, the hypoxic response could be mediated by HIF-2α, which is metabolically better than the HIF-1α-mediated response involved in the inflammatory response. In the SVF of eWAT treated with AX, the genes for adipose progenitor cells and cell markers constituting vasculature were elevated, suggesting the possibility of more histologically robust vascularization and stabilization of the progenitor cell niche.

While hypoxic response-mediated VEGF production in adipocytes contributes to metabolic homeostasis through angiogenesis and beigeing/briting [34], it also promotes macrophage infiltration into the AT [35]. Furthermore, the hypoxic response of infiltrated MΦ leads to the retention of chronic inflammation and pathological remodeling of AT [20].

The hypoxic response described above is based on the results of studies using HIF-1α whole body knockout or cell-specific knockout mice. Recently, it was shown from HIF-2α knockout mice and from studies using HIF-2α transgenic macrophages that HIF-2α in MΦ enhances anti-inflammatory effects and insulin sensitivity in contrast to HIF-1α in AT [30]. Relatedly, it was reported from the results of PHD2 (*Engl1*) knockout mice that pseudo-hypoxia induces a hypoxic response involving HIF-2α from HIF-1α- dominant hypoxic response. These knockout mice demonstrated healthy AT expansion and normal glucose tolerance in obesity [29]. Thus, in AT, HIF-1α and HIF-2α were shown to play contrasting roles in MΦ. In adipocytes, HIF-2α, similar to HIF-1α, plays an important role in fat beigeing and cold-induced thermogenesis. In addition, it plays an important role in atherosclerosis. As an important mechanism, HIF-2α enhances the degradation of ceramide through the gene induction of alkaline ceramidase 2 (ACER2) [36]. As a result, it promotes cold-induced thermogenesis and inhibits atherosclerosis. Thus, HIF-1 and HIF-2α possess opposing functions in inflammation, although there is some overlap in their functions. Especially in AT MΦ, the function of these two HIF isoforms in inflammation and insulin resistance can be described as yin and yang. There should be a consideration of HIFs to understand why these differences in function occur.

Perhaps these responses depend on the degree of oxygen tension in the region of the tissue and the type of HIF molecules that responds to hypoxia. Generally, under aerobic conditions, HIF-1/2α is usually hydroxylated by specific prolyl hydroxylases (PHDs) at conserved proline residues situated within the oxygen-dependent degradation domain (ODD) in a reaction, requiring molecular oxygen, 2-oxoglutarate, ascorbate, and Fe^2+^ as a cofactor. HIF-1/2α hydroxylation promotes the binding of von Hippel–Lindau protein (pVHL) to the HIF-1/2α ODD. pVHL forms the substrate recognition module of the E3 ubiquitin ligase complex and induces polyubiquitination and the proteasomal degradation of HIF-1/2α. In hypoxia, PHD activity is inhibited, pVHL binding is blocked, and HIF-1α and HIF-2α are stabilized. PDH activity is inhibited by ROSs, such as super oxide anion radical and nitric oxide (NO), and certain metal ions [27,28,37]. Therefore, the hypoxic response may also be enhanced by ROS. Although the mechanism of action of AX is obscure, it has been shown to inhibit the transfer of HIF-1α into the nucleus in the 7,12-dimethylbenz[a]anthracene (DMBA)-induced hamster buccal pouch (HBP) carcinogenesis model [38]. As a matter of course, AT is a tissue that stores fat and features a large amount of lipids. While the lipids provide energy, free saturated fatty acids act as mediators of inflammation. Similarly, unsaturated fatty acids are peroxidated by various ROS generated by NADPH oxidase and myeloperoxidase in immune cells, including monocytes such as macrophages. In addition to direct lipid peroxidation by ROS generated by these enzymes, a more potent peroxynitrite is generated by reaction with nitric oxide. Nitric oxide (NO) production is also increased by the gene expression of iNOS (NOS2) and subunit 4-2 of cytochrome c oxidase (COX4-2), which are targets of HIF-1α [28]. In addition to these well-known ROS peroxidation reactions, it has recently become clear that singlet oxygen, which was classically thought to be generated by photosensitization, is also generated in vivo by reactions with myeloperoxidase and decomposing of peroxynitrite under dark conditions [39]. Recently, it has also been reported that specific lipid peroxides modified by singlet oxygen increase in the plasma at a relatively early stage of type 2 diabetes or before the onset of the disease [40,41]. Thus, it is becoming clear that singlet oxygen is also closely related to insulin resistance [39]. As a result, the hypoxia-induced accumulation of M1 macrophages leads to synergistic cytotoxicity with the hypoxic response, and the DAMPs generated by the cytotoxicity further accumulate M1 macrophages, creating a vicious cycle including ROS-mediated oxidative stress. From these viewpoints, AX has a remarkable inhibitory effect on the progression of the fat peroxidation chain reaction and on the direct peroxidation of lipids by singlet oxygen [42,43]. It has also been reported that AX inhibits the activation of the classical NFκB pathway through various cytokines and NO stimulation in the strain of monocytes, RAW264.7, and thereby suppresses excessive inflammatory responses [44]. In this study, the accumulation of AX was observed in the AT of mice at a level that could be visually confirmed, suggesting that AX is close to the actual effective dose.

It is necessary to consider HIF-2α rather than HIF-1α, as we found a significant difference in the expression levels of HIF-2α in this study. The effects of PHDs on different HIF isoforms are generally not equivalent, with PHD2 affecting HIF-1α more than HIF-2α and PHD3 (*Egln3*) affecting HIF-2α more than HIF-1α. In addition, HIF-2α is hydroxylated by PHDs at a much lower efficiency than HIF-1α, resulting in the stable activation of HIF-2α at higher oxygen concentrations than HIF-1α [27,45].

A further possible mechanism of action is that in skeletal muscle, HIF-2α may cooperate with PGC-1α to promote muscle fiber switches, including capillary formation, which may also occur in AT [46]. We previously found that AX induces AMPKα1/2-mediated expression of PGC-1α in skeletal muscle and concomitant enhancement of vascular marker genes [10]. In this case, induction was also observed in HIF-2α. Other authors have also shown, in a model of skeletal muscle disuse atrophy, that AX intake can prevent ROS-mediated skeletal muscle atrophy and also prevent capillary regression [47]. They also demonstrated that AX suppressed vascular regression in a UV-induced skin aging model using hairless mice [48]. In addition to this, our previous reports and the findings of the present study indicate that the gene expression of ERRα and ERRγ is upregulated by the administration of AX in skeletal muscle and eWAT [10]. It is known that the activation of AMPKα2 leads to the induction not of only PGC-1α but also ERRα gene expression [49]. Genetic knockouts of ERRα and ERRγ impair tissue and vascular regeneration during skeletal muscle injury, and these genes are also known to play an important role in the formation of vasculatures [50,51]. It is still unclear whether this AX-induced enhancement in the gene expression of ERRs were caused by AX acting as some kind of ligand or as a result of AMPK activation, thus requiring further study.

The reason why the results of the AX intervention study for obesity with already established pathology were more partial than those of the preventative administration was probably because pathological remodeling was already developed in AT and the niches of progenitor cells necessary for the healthy adipogenesis were already exhausted. Consistent with this observation, eWATs in the control group decreased in weight from week 10 to week 24, and they were more atrophic. However, it is possible to ameliorate glucose intolerance caused by saturated free fatty acids by consuming fat in skeletal muscle or partially in the liver with AX intervention. In this study, the obese mice did not receive any exercise intervention. We previously demonstrated that the administration of AX is beneficial for improving metabolic parameters in obesity in addition to “daily exercise” [10]. Therefore, in the case of use in humans, it is also preferable to intervene as early as possible in obesity. Since this was complementary to exercise therapy, AX interventions should be used in combination with exercise.

Our results indicate that AX suppresses ectopic fat in other tissues by maintaining tissue integrity in AT by protecting the microenvironment of progenitor cells and stem cells from inflammation. These results suggest that the supplementation of AX may be beneficial for preventing not only metabolic syndrome, but also cardiovascular disease, liver cirrhosis, liver cancer, diabetes and its complications, dementia, and other insulin resistance-related cardio-metabolic complications. AX is considered a safe and viable natural dietary source and it could be used for the treatment of various disorders. From the perspective of maintaining the microenvironment, it may be useful not only for anti-aging but also for the prevention of diseases that have not received much interest before. Further studies are warranted to clarify the clinical implications of AX.

## 5. Conclusions

We conclude that AX treatment reduces oxidative stress and macrophage infiltration into AT, maintains APs and MSC niches, prevents pathological AT remodeling, and improves AT functions. In summary, AX is a natural compound that maintains the metabolic tolerance and integrity of AT and contributes to its healthy functions.

## Figures and Tables

**Figure 1 nutrients-13-04374-f001:**
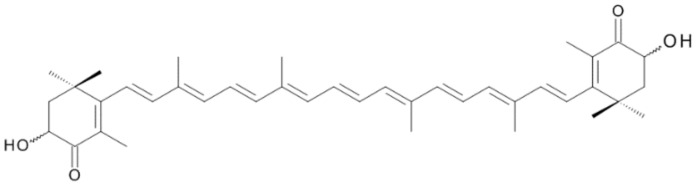
Chemical structure of Astaxanthin (AX).

**Figure 2 nutrients-13-04374-f002:**
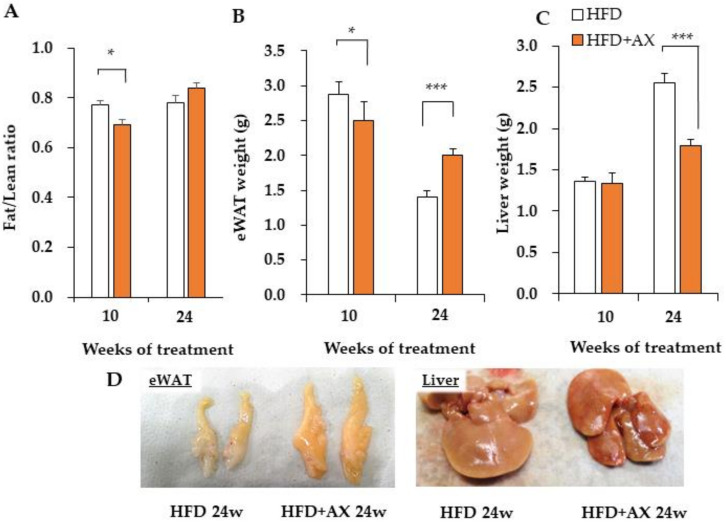
AX administration changed the body composition and tissue weight of high-fat diet-loaded (HFD) mice. (**A**) Body composition by NMR, Fat/lean ratio, epididymal adipose tissue (eWAT) weight (**B**), and liver weight (**C**) of control mice on a high-fat diet and AX-treated mice at 10 and 24 weeks. (**D**) Representative images of eWAT and liver tissues. (*n* = 5–6 per group). All values are represented as means ± S.E.M. * *p* < 0.05, *** *p* < 0.001 (HFD vs. HFD + AX). Statistical tests were performed as follows: two-way repeated-measures ANOVA, a post-hoc Tukey-Kramer for each point.

**Figure 3 nutrients-13-04374-f003:**
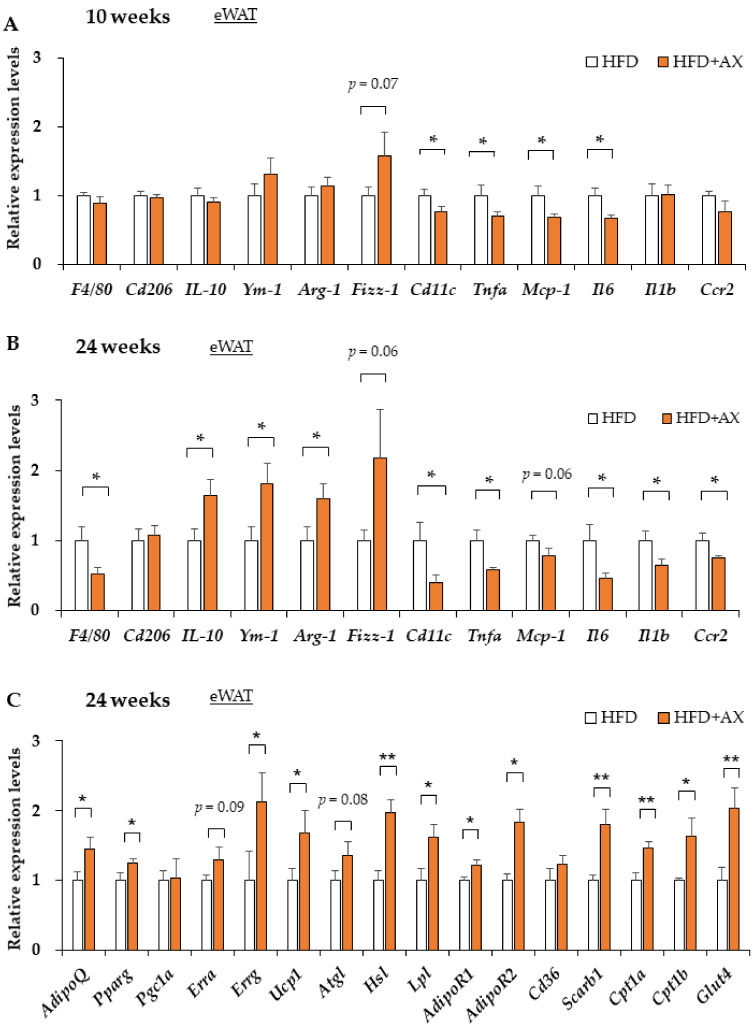
AX administration regulated the gene expression of pro-inflammatory markers and metabolic markers in the eWAT compared to HFD-treated control mice. Gene expression of pro-inflammatory and anti-inflammatory-related marker genes in eWAT of HFD-fed mice either 8 weeks (**A**) or 24 weeks (**B**) after AX administration. Gene expression of metabolism markers, including adipokines, lipolysis and energy metabolisms in eWAT of HFD treated for 10 weeks (*n* = 5–6 per group) (**C**). All values are presented as the means ± S.E.M. * *p* < 0.05, ** *p* < 0.01 (HFD vs. HFD + AX). Statistical analysis was performed using Student’s *t*-test.

**Figure 4 nutrients-13-04374-f004:**
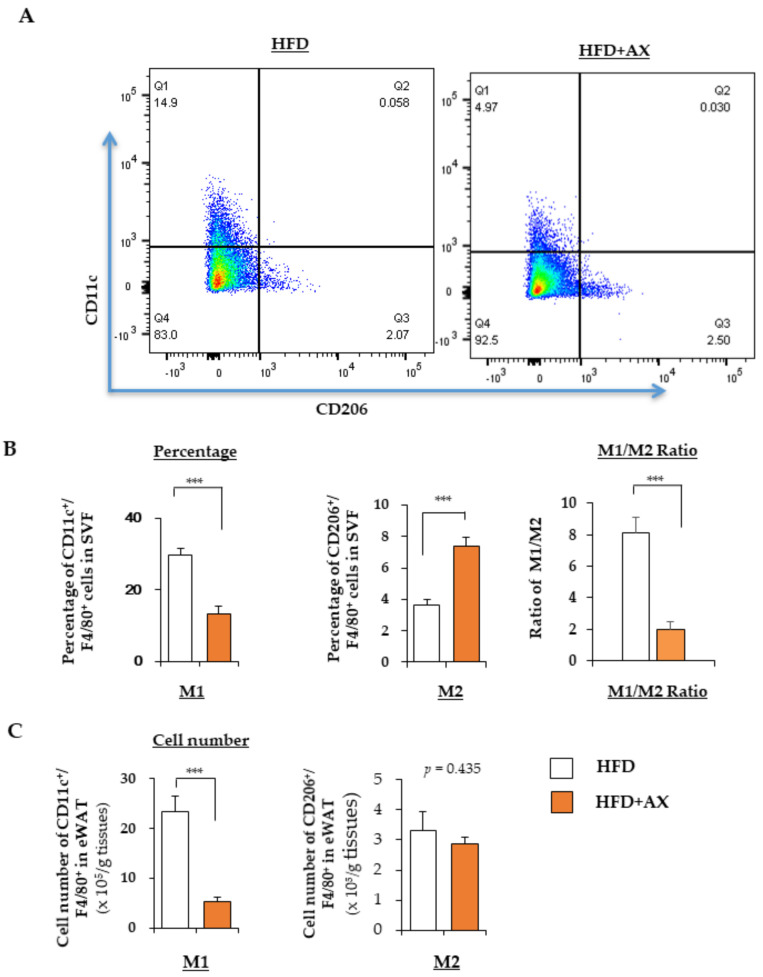
AX administration attenuated the infiltration of M1 macrophages (MΦ) into adipose tissue of HFD-fed mice. (**A**) Representative flow cytometry analysis of immune cells in eWAT of AX-treated HFD and HFD control mice (*n* = 6 mice/group). For this, the live cells were gated for CD45+ cells, followed by F4/80^+^ MΦ, and CD206^+^ M2 MΦ, and CD11c^+^ M2 MΦ. The percentages, their M1/M2 MΦ ratio (**B**) and the cell numbers (**C**) of M1 (CD11c^+^) and M2 (CD206^+^) MΦ relative to F4/80^+^ cells in the stromal vascular fraction (SVF) of eWAT from mice treated with HFD for 24 weeks. All values are presented as the means ± S.E.M. *** *p* < 0.001 (HFD vs. HFD + AX). Statistical analysis was performed using Student’s *t*-test.

**Figure 5 nutrients-13-04374-f005:**
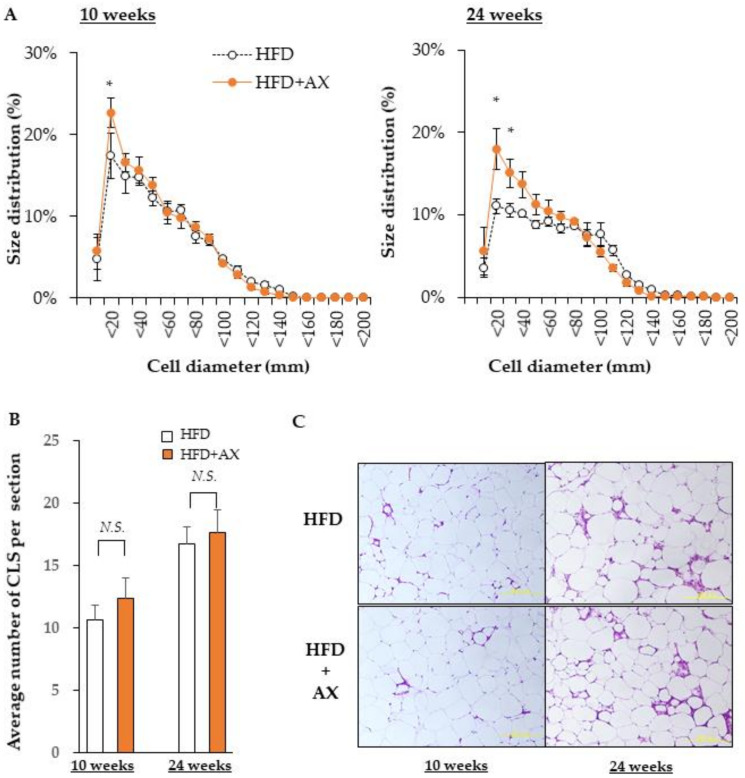
AX administration significantly altered the size of adipocytes in eWAT. (**A**) Diameter of adipocytes calculated from the area of adipocytes. (**B**) Average numbers of the crown like structures in each Hematoxylin and Eosin (H&E) stained section. (**C**) Representative H&E stained histological sections (*n* = 3–4/each group, each sample was measured in at least four sections). All values are represented as means ± S.E.M. * *p* < 0.05, *N.S.*: not significant (HFD vs. HFD + AX). Statistical tests were performed as follows: (**A**) two-way repeated-measures ANOVA, a post-hoc Dunnet’s-test for each point. (**B**) Student’s *t*-test.

**Figure 6 nutrients-13-04374-f006:**
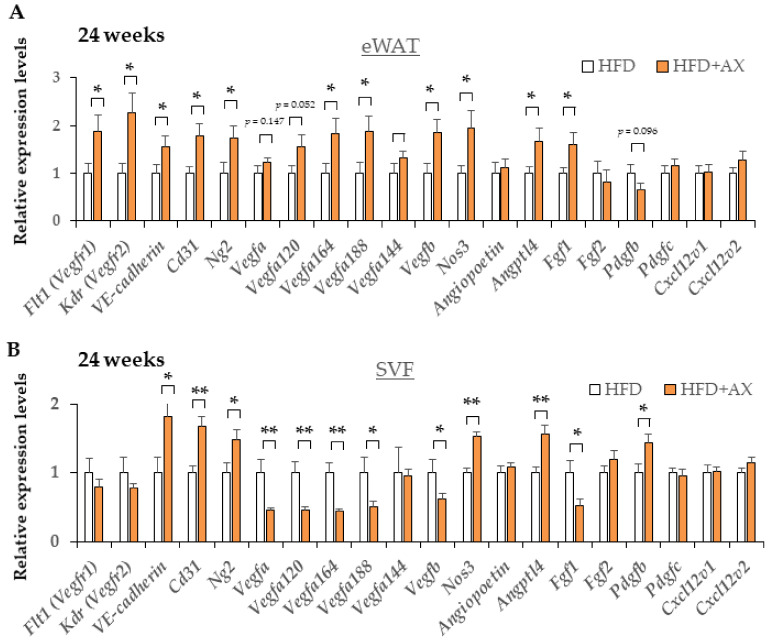
AX administration partially upregulated the gene expression of vascularization markers in eWAT and their SVF compared to HFD-treated control mice for 24 weeks. Gene expression of vascular marker genes in eWAT (**A**) and their SVF (**B**) (*n* = 5–6/each group). All values are represented as means ± S.E.M. * *p* < 0.05, ** *p* < 0.01 (HFD vs. HFD + AX). Statistical tests were performed as follows: Student’s *t*-test.

**Figure 7 nutrients-13-04374-f007:**
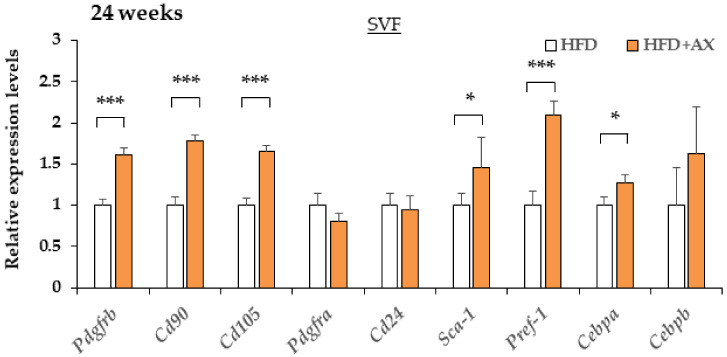
AX administration altered the gene expression of adipocyte progenitor cells, and mesenchymal stem cell markers in SVF compared to HFD-treated control mice for 24 weeks. Gene expression of adipose progenitor cells, and stem cell marker genes in SVF from eWAT of mice treated HFD for 24 weeks. (*n* = 5–6/each group). All values are represented as means ± S.E.M. * *p* < 0.05, *** *p* < 0.001 (HFD vs. HFD + AX). Statistical tests were performed as follows: Student’s *t*-test.

**Figure 8 nutrients-13-04374-f008:**
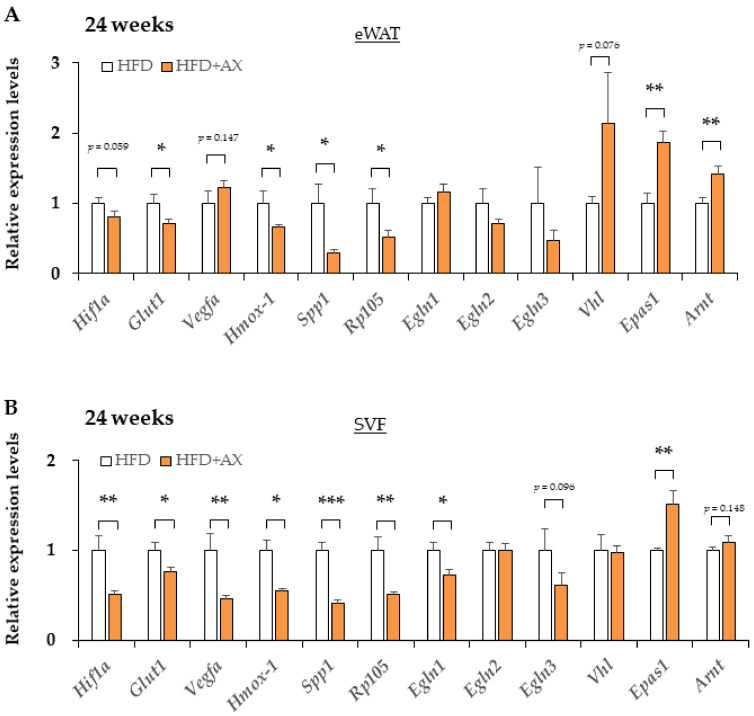
The hypoxic response altered from HIF-1 to HIF-2 dominant in eWAT and its SVF of AX-treated HFD mice compared to HFD control mice for 24 weeks. Gene expression of vascular marker genes in eWAT (**A**) and their SVF (**B**). (*n* = 5–6/each group). All values are represented as means ± S.E.M. * *p* < 0.05, ** *p* < 0.01, *** *p* < 0.001 (HFD vs. HFD+AX). Statistical tests were performed as follows: Student’s *t*-test.

## Data Availability

All the data underlying the results are available as part of the article and no additional source data are required.

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
