# Peer review of "Astaxanthin, a Marine Carotenoid, Maintains the Tolerance and Integrity of Adipose Tissue and Contributes to Its Healthy Functions"

_nutrients, 2021, doi:10.3390/nu13124374_

Round 1

Reviewer 1 Report

The authors in this study highlighted that astaxanthin improves obesity-induced glucose intolerance and insulin resistance. This demonstrates how medicinal plants are a source of nutrients useful for health

The manuscript is well organized although

Even if the authors used a pure compound, they should be more precise and indicate the Plants in which astaxanthin is present

Also could they indicate the purity of the compound?

I suggest adding these manuscripts (see below) in the introduction to emphasize the importance of medicinal plants and their potential help for people's health.

Fernández J, Silván B, Entrialgo-Cadierno R,Villar  C.J, Capasso R., Uranga J. A., Lombó F, l Abalo R., Antiproliferative and palliative activity of flavonoids in colorectal cancer. Biomed Pharmacother. 2021; 143: 112241, doi.org/10.1016/j.biopha.2021.112241

Küpeli Akkol E, Genç Y, Karpuz B, Sobarzo-Sánchez E, Capasso R. Coumarins and

Coumarin-Related Compounds in Pharmacotherapy of Cancer. Cancers (Basel). 2020

Jul 19;12(7):1959.

AÄŸagündüz D, Çelik MN, Çıtar DazıroÄŸlu ME, Capasso R. Emergent Drug and

Nutrition Interactions in COVID-19: A Comprehensive Narrative Review. Nutrients.

2021 May 4;13(5):1550

Do the authors think the microbiota is involved?

In the Discussion, the Authors should highlight the possible clinical significance

Author Response

Thank you very much for evaluating our manuscript. We have tried our best to address all the queries. Please find below the response of all queries.

Response to Reviewer 1 Comments

The authors in this study highlighted that astaxanthin improves obesity-induced glucose intolerance and insulin resistance. This demonstrates how medicinal plants are a source of nutrients useful for health.

The manuscript is well organized although.

Point 1; Even if the authors used a pure compound, they should be more precise and indicate the Plants in which astaxanthin is present. Also could they indicate the purity of the compound?

Response

Thank you for the comment. For our study, we used a powder derived from Haematococcus algae, which contains 2% of AX. Accordingly, we have added the following text in the Materials and Methods section. “commercially available astaxanthin (AX) powder was purchased from Fuji Chemical Industries USA; Inc. (Product name; P2AF, containing 2% of AX from Haematococcus pluvialis, Burlington, NJ)”

Point 2; I suggest adding these manuscripts (see below) in the introduction to emphasize the importance of medicinal plants and their potential help for people's health.

Fernández J, Silván B, Entrialgo-Cadierno R,Villar  C.J, Capasso R., Uranga J. A., Lombó F, l Abalo R., Antiproliferative and palliative activity of flavonoids in colorectal cancer. Biomed Pharmacother. 2021; 143: 112241, doi.org/10.1016/j.biopha.2021.112241

Küpeli Akkol E, Genç Y, Karpuz B, Sobarzo-Sánchez E, Capasso R. Coumarins and Coumarin-Related Compounds in Pharmacotherapy of Cancer. Cancers (Basel). 2020 Jul 19;12(7):1959.

AÄŸagündüz D, Çelik MN, Çıtar DazıroÄŸlu ME, Capasso R. Emergent Drug and Nutrition Interactions in COVID-19: A Comprehensive Narrative Review. Nutrients. 2021 May 4;13(5):1550

Response

Thank you for valuable suggestion. We have cited the reference wherever possible.

Point 3; Do the authors think the microbiota is involved?

Response

Thank you for the comment. We think microbiota is not involved. This possibility was also initially investigated with regard to the see the effects of AX on gut microbiota. Our preliminary real-time PCR evaluation showed no changes in Akkermansia muciniphila or butyrate-producing bacteria, which are beneficial for energy metabolism and inflammation, compared to the control group. Only the Firmicutes and the Lactobacillus group contained in them changed. We could not visually identify any inflammation or morphological changes in the colon or cecum. Therefore, under the conditions of our study, we considered that the effects of intestinal microflora could not be linked to known reports and did not conduct further research.

We acknowledge that the gut microbiota plays an important role in systemic inflammation and glucose metabolism. We have reported several studies on the relationship between gut microbiota and energy metabolism.

https://www.nature.com/articles/s41598-020-62506-w

https://doi.org/10.1016/j.isci.2021.102445

https://www.jci.org/articles/view/86674

There are a few patents and article of changes in the intestinal microflora with AX from Haematococcus algae.

https://patents.google.com/patent/WO2014208511A1/en

https://patents.google.com/patent/JP2019131527A/en

https://doi.org/10.1093/jn/nxaa222

Point 4; In the Discussion, the Authors should highlight the possible clinical significance

Response

Thank you for the valuable suggestions. We have included the following paragraph at the end of the discussion

Line591

“Our results indicate that AX suppresses ectopic fat in other tissues by maintaining tissue integrity in adipose tissue by protecting the microenvironment of progenitor cells and stem cells from inflammation. These results suggest that aggressive consumption of AX may be beneficial in preventing not only metabolic syndrome, but also cardiovascular disease, liver cirrhosis and liver cancer, diabetes, diabetes and its complications, dementia, and other insulin resistance-related metabolic complications. From the aspect of maintaining the microenvironment, it may be useful not only for anti-aging but also for the prevention of diseases that have not received much interest before. Further studies are warranted to clarify clinical implications of AX”.

Reviewer 2 Report

In this manuscript seems to be deepened the effects of astaxanthin on the adipocytes. However, there are too many citations to a previously published word (reference 10) that did not enable to find exactly what is new in this manuscript. Maybe should be emphasized better the main findings of this manuscript.

There are some other points that should ameliorated:

Abstract: the abstract did not reflect the content of the manuscript (in the studies were used animals but in the abstract nothing is referred about this and also about the methods employed, etc)

The material and methods section should be the last one before conclusions (otherwise the supplementary figure S8 cited in line 113 was in the incorrect order)

Line 161: Oxidative stress analysis was performed according to the manufacturer's instructions (see ref [10]), but I did not find the manufacturer instruction in the ref 10. Please correct.

Regarding the supplementary figure S1 there is an error as the authors mentioned that was also presented in ref 10 but in the legend put ref. 5.

Author Response

Thank you very much evaluating our manuscript. We have tried our best to address all the queries. Please find below the response of all queries.

Response to Reviewer 2 Comments

Point 1; In this manuscript seems to be deepened the effects of astaxanthin on the adipocytes. However, there are too many citations to a previously published word (reference 10) that did not enable to find exactly what is new in this manuscript. Maybe should be emphasized better the main findings of this manuscript.

Response

Thank you for the comment. We agree with reviewer and we refrain to repeat same reference.

Point 2; There are some other points that should ameliorated:

Abstract: the abstract did not reflect the content of the manuscript (in the studies were used animals but in the abstract nothing is referred about this and also about the methods employed, etc)

Response

Thank you for pointing that out. We have included the following sentence in the abstract. Line36 “We fed 6-week-old male C57BL/6J on high-fat-diet (HFD) supplemented with or without AX for 24 weeks. We determined the effect of AX at 10 and 24 weeks of HFD with or without AX on various parameters including insulin sensitivity, glucose tolerance, inflammation, and mitochondrial function in adipose tissue. We found that AX significantly reduced oxidative stress and macrophage infiltration into adipose tissue, as well as maintaining healthy adipose tissue function.” Also, to clarify the focus, the sentence before the added sentence was deleted.

Point 3; The material and methods section should be the last one before conclusions (otherwise the supplementary figure S8 cited in line 113 was in the incorrect order)

Response

Thank you for pointing that out.

We have deleted the description of Supplementary Fig. S8 from the Materials and methods section. Instead, we added description to the legend of Supplementary Fig S8.

Point 4; Line 161: Oxidative stress analysis was performed according to the manufacturer's instructions (see ref [10]), but I did not find the manufacturer instruction in the ref 10. Please correct.

Response

Thank you for pointing that out. We have removed the citation for reference 10. We have included the kit official product name and product number.

Point 5; Regarding the supplementary figure S1 there is an error as the authors mentioned that was also presented in ref 10 but in the legend put ref. 5.

Response

Thank you for pointing that out. We have corrected

Round 2

Reviewer 2 Report

In general, the changes required were performed.